

# Characterizing abnormal behavior in a large population of zoo-housed chimpanzees: prevalence and potential influencing factors

Sarah L. Jacobson[1], Stephen R. Ross[1] and Mollie A. Bloomsmith[2]

[1] Lester E. Fisher Center for the Study and Conservation of Apes, Lincoln Park Zoo, Chicago, Illinois, United States
[2] Animal Resources, Yerkes National Primate Research Center, Atlanta, Georgia, United States

Corresponding author
Stephen R. Ross, sross@lpzoo.org

## ABSTRACT

Abnormal behaviors in captive animals are generally defined as behaviors that are atypical for the species and are often considered to be indicators of poor welfare. Although some abnormal behaviors have been empirically linked to conditions related to elevated stress and compromised welfare in primates, others have little or no evidence on which to base such a relationship. The objective of this study was to investigate a recent claim that abnormal behavior is endemic in the captive population by surveying a broad sample of chimpanzees (*Pan troglodytes*), while also considering factors associated with the origins of these behaviors. We surveyed animal care staff from 26 accredited zoos to assess the prevalence of abnormal behavior in a large sample of chimpanzees in the United States for which we had information on origin and rearing history. Our results demonstrated that 64% of this sample was reported to engage in some form of abnormal behavior in the past two years and 48% of chimpanzees engaged in abnormal behavior other than coprophagy. Logistic regression models were used to analyze the historical variables that best predicted the occurrence of all abnormal behavior, any abnormal behavior that was not coprophagy, and coprophagy. Rearing had opposing effects on the occurrence of coprophagy and the other abnormal behaviors such that mother-reared individuals were more likely to perform coprophagy, whereas non-mother-reared individuals were more likely to perform other abnormal behaviors. These results support the assertion that coprophagy may be classified separately when assessing abnormal behavior and the welfare of captive chimpanzees. This robust evaluation of the prevalence of abnormal behavior in our sample from the U.S. zoo population also demonstrates the importance of considering the contribution of historical variables to present behavior, in order to better understand the causes of these behaviors and any potential relationship to psychological wellbeing.

## INTRODUCTION

Abnormal behaviors in primates have been generally defined as behaviors that are atypical of the species and/or occur at different frequencies in captivity than in the wild

(*Erwin & Deni, 1979*; *Walsh, Bramblett & Alford, 1982*). These behaviors can result from both proximate and past exposure to chronic aversive stimuli, including environments that limit the ability to perform species-typical behaviors, repeated stressful procedures such as sedations for clinical procedures (*Lutz, Well & Novak, 2003*), and/or atypical early social experiences such as reduced or absent maternal care (*Brent, Lee & Eichberg, 1989*; *Kalcher et al., 2008*; *Freeman & Ross, 2014*). Other intrinsic factors such as sex, species, and animal temperament (*Vandeleest, McCowan & Capitanio, 2011*; *Gottlieb, Capitanio & McCowan, 2013*) can also influence the expression of some abnormal behaviors. In part because of the association with suboptimal social and physical environments, abnormal behaviors are often considered to be reliable indicators of psychological distress and as such, poor welfare (*Mason, 1991*; *Garner, 2005*). Studying the prevalence and persistence of these behaviors in captive environments is critical to better understanding the factors contributing to the wellbeing of captive primates.

Despite its importance to improving the captive management of primates, the study of abnormal behavior can be quite challenging for several reasons. The first is a lack of consistency in the types and definitions of behaviors considered to be atypical. *Walsh, Bramblett & Alford (1982)* developed one of the most widely-used ethograms of abnormal behaviors in chimpanzees (*Pan troglodytes*), but there are a variety of others for this species that differ in important ways (see *Nash et al., 1999*; *Hook et al., 2002*; *Birkett & Newton-Fisher, 2011*). Some classifications distinguish between behaviors that are pathological and harmful (e.g. self-injurious behavior) and those that are less severe (e.g. repetitive motions) (*Bayne & Novak, 1998*). Other investigators have not made such a distinction, treating a variety of abnormal behaviors as functionally equivalent in terms of impact on the animal (*Birkett & Newton-Fisher, 2011*). A second challenge relates to the interpretation of the context in which particular behaviors are performed. In some cases, behaviors can occur in both a species-typical context as well as those that may be more likely tied to an underlying distressful state. For example, some researchers have recognized this difficulty and advised that caution be used when categorizing rocking as an abnormal behavior because of the difficulties in distinguishing instances when the behavior is communicative in chimpanzees (*Fritz et al., 1992*; *Ross & Bloomsmith, 2011*). These types of definitional incongruities can lead to difficulties interpreting and comparing results of abnormal behavior studies.

Perhaps the broadest challenge related to the study of abnormal behavior is related to the interpretation of behaviors and their potential underlying etiologies. Although some behaviors have been empirically linked to conditions related to elevated stress and compromised welfare, others have little or no evidence on which to base such a relationship. Many stereotypies are commonly used as indicators of reduced wellbeing, but this relationship is convoluted due to the complex mechanisms underlying stereotypic behaviors. *Mason & Latham (2004)* suggest that some stereotypies can develop as coping mechanisms for animals and with repetition shift into an automatic behavior that is not necessarily reflective of their current environment. Other stereotypies may even demonstrate a degree of behavioral flexibility as animals attempt to satisfy a motivation to perform a natural behavior pattern in a captive environment (*Mason & Latham, 2004*).

Additionally, the relationship between stereotypy and self-injurious behavior is poorly understood, as some self-directed stereotypies can cause physical injury while the same behavior can also occur without injury. The classification of depilation and regurgitation and reingestion as problematic behaviors is also debated (*Baker & Easley, 1996*; *Hosey & Skyner, 2007*). *Novak et al. (2006)* advocated for further study of the biological bases of these categories of behavior to determine whether they represent different manifestations of the same underlying mechanism. Clearly, a better understanding of the etiologies of those behaviors classified as abnormal will be critical to determining which can be used as reliable indicators of negative welfare in captive primates.

One of the most comprehensively studied factors influencing the development of abnormal behaviors is early social experience and in particular, the rearing history of captive primates. Early maternal separation in captive macaques and chimpanzees has repeatedly been shown to lead to stereotypies, self-injurious behavior, and incompetent social and reproductive behavior (*Harlow & Harlow, 1965*; *Rogers & Davenport, 1969*; *Fritz et al., 1992*; *Nash et al., 1999*; *Martin, 2002*). Stress physiology has also supported this connection, demonstrating a dysregulation of the hypothalamic-pituitary-adrenocortical axis in primates that are separated from their mothers early in life, along with elevated expression of abnormal behaviors (*Feng et al., 2012*). In contrast, some abnormal behaviors in chimpanzees, such as coprophagy, appear to be more prevalent in mother-reared individuals compared to those raised by humans (*Nash et al., 1999*; *Bloomsmith et al., 2006*). Although not all primates who have experienced maternal separation demonstrate behavioral abnormalities, their rearing history appears to be an important factor influencing abnormal behavior.

The physical environment experienced by captive primates may also play a role in the development of abnormal behaviors, as the absence of appropriate sensory and motor stimulation can lead to stereotypies (*Berkson, Mason & Saxon, 1963*; *Harlow & Harlow, 1965*). When captive environments fail to provide adequate opportunities for natural behavior patterns, some atypical behaviors such as rocking or pacing may develop as a form of self-stimulation (*Walsh, Bramblett & Alford, 1982*). Further evidence for the importance of physical environments comes from studies that demonstrate reductions in abnormal behaviors following a move to a more natural or enriched environment (*Pfeiffer & Koebner, 1978*; *Brent, Lee & Eichberg, 1989*; *Schapiro & Bloomsmith, 1994*; *Novak et al., 1998*; *Ross et al., 2011*). Given this broad support for a link between a primate's current and past physical and social environments, it seems imperative to consider a wide range of potential influencing factors when investigating the prevalence of abnormal behaviors.

In this study, we surveyed animal care staff to assess the prevalence (proportion of individuals who exhibited a behavior at least once in a two-year period) of abnormal behaviors in a broad sample of chimpanzees living in accredited zoological parks. We also evaluated potential links between historical variables and abnormal behaviors. Although many primates exhibit abnormal behavior, chimpanzees are a particularly relevant species with which to investigate these phenomena. Chimpanzees are a socially and cognitively complex species and they perform a variety of abnormal behaviors in a variety of settings

(*Nash et al., 1999*; *Birkett & Newton-Fisher, 2011*). Indeed, the captive chimpanzee population in North America provides a unique opportunity to investigate factors influencing the development of abnormal behavior, in part because of the inherent range of rearing and housing conditions experienced by these individuals. Chimpanzees living in accredited zoos come from a broad diversity of backgrounds including those born and raised in zoos, those born in the wild and imported (many years ago) to North America, and those born in other captive settings such as research laboratories or those privately owned as pets or performers, and later moved to zoos.

A recent study of abnormal behavior in chimpanzees attempted to quantify the prevalence and diversity of abnormal behaviors in the zoo setting and concluded that "abnormal behaviour is endemic in captivity" (*Birkett & Newton-Fisher, 2011*: 5). The study included 40 chimpanzees at only six institutions to arrive at this conclusion. Here, we use a broader sampling of the zoo-housed chimpanzee population to assess the prevalence of these behaviors, surveying animal staff at 26 accredited zoos to collect data on 165 chimpanzees. Furthermore, we used information on their origin and rearing histories to investigate the factors influencing the expression of abnormal behaviors. Our objective was to characterize these populations and gather information to help influence future management practices.

## METHODS

### Data collection

To assess the prevalence of abnormal behavior of a large sample of chimpanzees housed in accredited zoos in the United States, we surveyed animal care staff who were familiar with those individuals. Using *PMCTrack software (2016)*, we administered an online questionnaire to all Institutional Representatives of the Chimpanzee Species Survival Plan (SSP). These individuals included curators, managers and zookeepers who worked regularly with the chimpanzees and were responsible for their care and management. Respondents were asked to note, for each individual chimpanzee at their institution, which abnormal behaviors were displayed at least once in the previous two-year period. This is an inherently conservative approach that is more likely to *overestimate* the prevalence of abnormal behaviors than to underestimate it because even highly infrequent behaviors could be recorded. The list of abnormal behavior categories (Table 1) was developed in part from *Birkett & Newton-Fisher (2011)* to facilitate comparisons with the results of that study.

### Historical variables

Information about the chimpanzees, including their sex, rearing, and origin was drawn from the North American Regional Studbook for Chimpanzees (*Ross, 2015*). Rearing was simply categorized as mother-reared or non-mother-reared as many historical records did not provide sufficient information to be more precise about rearing conditions. The origin of individuals was defined as the location at which the chimpanzee was born, and therefore the environment where it spent at least some of its early developmental period. Origins were categorized as zoo, laboratory, wild, and private. The private designation

**Table 1 The definitions of abnormal behavior categories for chimpanzees used in this study.**

| | |
|---|---|
| Coprophagy | Ingestion of feces |
| Hair pluck | Pulling out hair on self or another |
| Rock | Repetitive and sustained swaying movement without piloerection |
| Regurgitation & Reingestion | The deliberate regurgitation of food and subsequent consumption of the food |
| Self-injurious behavior | Biting, picking, or scratching at own body to cause injury |
| Pacing | Locomoting repetitively along the same path with no clear objective |
| Other | Any other behavior deemed abnormal, space to describe |

included chimpanzees that were kept as pets or performers and therefore had significant human interaction during their development (*Freeman & Ross, 2014*).

## Facilities and subjects

We received surveys on 181 chimpanzees, but not all were complete. After removing incomplete surveys and excluding chimpanzees with unknown historical variables, our study sample consisted of 165 chimpanzees (see Table 2 for breakdown of independent variables). Chimpanzees ranged in age from 2–78 years old and age category distributions were comparable to those of the overall Association of Zoos and Aquariums (AZA) population with 10% immature (< 11), 65% adult (11–40), and 25% elderly (> 40) (*Ross, 2015*). They were all socially housed at one of 26 zoos accredited by the AZA. Though there was some variability in physical environments and management practices, all individuals lived under the regulatory framework provided by AZA and care was guided by the principles in the AZA Chimpanzee Care Manual (*AZA Ape Tag, 2010*). The sample was 38% male and 62% female, which matches the overall sex distribution of the entire AZA population (*Ross, 2015*).

## Analysis

We calculated the proportion of the sample who were reported to engage in any form of abnormal behavior, hereafter ABN-ALL. Additionally, due to the reported ambiguity of coprophagy as a reliable indicator of welfare (*Nash et al., 1999*; *Hopper, Freeman & Ross, 2016*), we calculated both the proportion of the sample that were reported to engage in coprophagy specifically (ABN-C), as well as the proportion who were reported to engage in any form of abnormal behavior *except* for coprophagy (ABN-XC).

The association between the historical variables and sex with the prevalence of abnormal behavior was assessed using binary logistic regression modeling. The variable of sex was considered in this analysis due to inconsistent results evaluating its association with abnormal behavior in past studies (*Fritz et al., 1992*; *Nash et al., 1999*). Analyses were conducted in R (*R Core Team, 2015*). The reference variable for sex was female, for rearing was mother-reared, and for origin was wild. A separate model was run for each of the dependent variables: ABN-ALL, ABN-C and ABN-XC. For all analyses an alpha value of $p \leq 0.05$ was considered significant.

**Table 2 Chimpanzees in study sample with each (A) Rearing history, (B) Origin category.**

**(A)**

| Rearing | Number of individuals | Percentage of sample |
|---|---|---|
| Mother-reared | 107 | 65% |
| Non-mother-reared | 58 | 35% |

**(B)**

| Origin | Number of individuals | Percentage of sample |
|---|---|---|
| Laboratory | 15 | 9% |
| Private | 15 | 9% |
| Wild | 34 | 21% |
| Zoo | 101 | 61% |

## RESULTS

Our survey indicated that 64% of the 165 chimpanzees had been seen to exhibit some form of abnormal behavior (ABN-ALL) at least once in the previous two years. Coprophagy (ABN-C) was the most prevalent abnormal behavior, with 41% of chimpanzees reported to engage in the behavior. Hair plucking was also fairly common, with 32% of the sample reported to engage in this behavior. Other behaviors were far less commonly reported (see Fig. 1). When removing chimpanzees that only exhibited coprophagy, 48% of the 165 chimpanzees exhibited abnormal behavior (ABN-XC).

Logistic regression models were created using all combinations of the three historical variables (sex, rearing, and origin) to determine which combination of these best predicted each of the dependent variables: ABN-ALL, ABN-C and ABN-XC. Only main effects were included in these models because all of the combinations of variables did not exist in our dataset, which compromised our ability to assess interaction effects. The best fit model was chosen through AIC comparison for each dependent variable (*Symonds & Moussalli, 2011*). When multiple models were statistically equivalent, the model that included the most variables was chosen in order to assess the influence of more factors on the chimpanzees' behavior. The best fit model for ABN-ALL included sex and rearing (not origin) as predictor variables and was statistically significant, $X^2(2) = 6.49$, $p = 0.04$. Table 3 reports the regression coefficients and the odds ratios for the model. The model explained 5.3% (Nagelkerke $R^2$) of the variance in ABN-ALL and correctly classified 61.8% of cases. The variable sex had a negative relationship with ABN-ALL such that male chimpanzees were 2.04 times less likely to exhibit abnormal behavior than females. Rearing was not a significant predictor of ABN-ALL in this model.

The best fit model for ABN-XC included all three predictor variables and was statistically significant ($X^2(5) = 19.18$, $p < 0.01$). Table 4 reports the regression coefficients and odds ratios for this model. The model explained 14.6% (Nagelkerke $R^2$) of the variance in ABN-XC and correctly classified 64.2% of cases. Rearing had a positive relationship with ABN-XC such that chimpanzees who were not mother-reared were

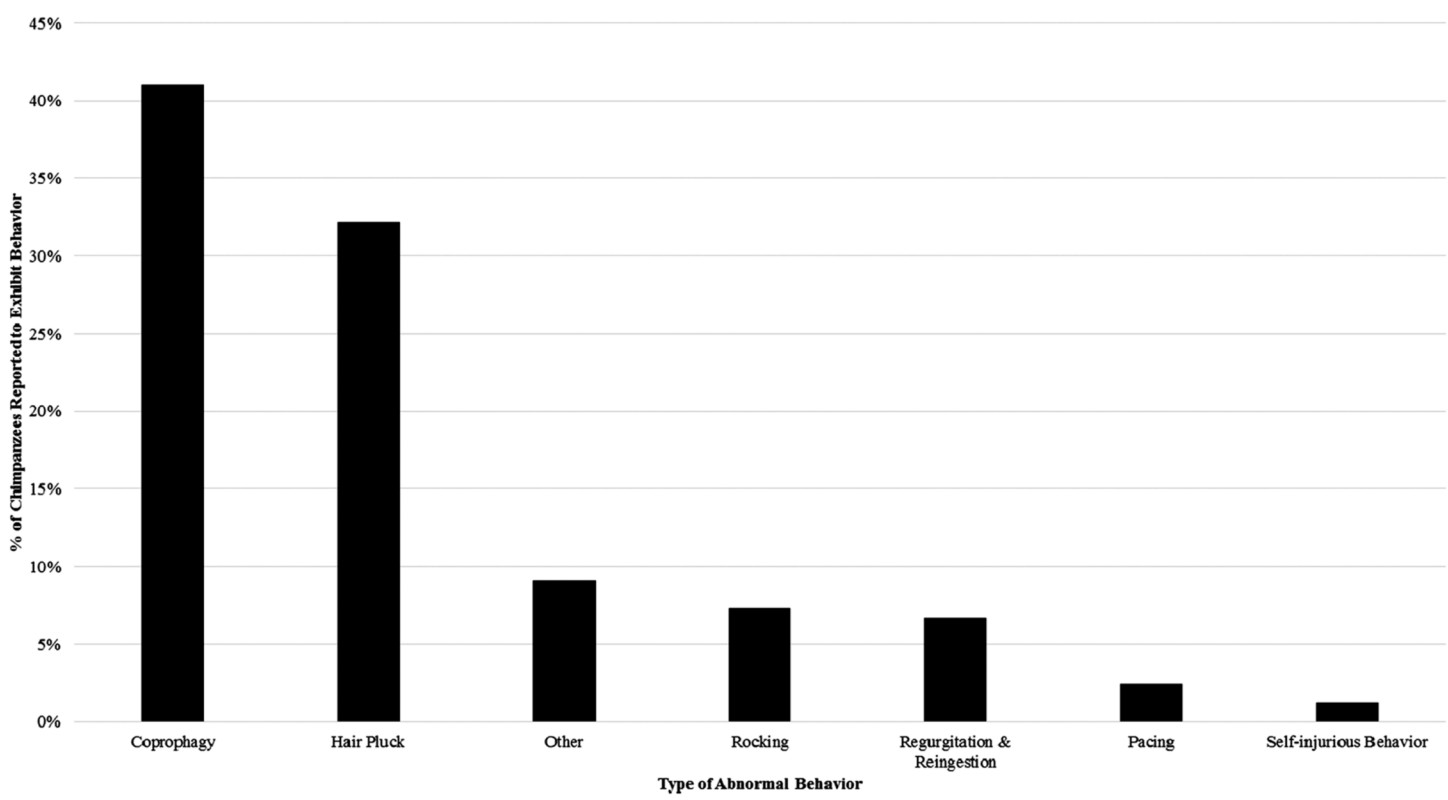

**Figure 1 The percentage of the study sample reported to engage in each category of abnormal behavior at least once from 2011–2013.**

**Table 3 Logistic regression model for any abnormal behavior (ABN-ALL) with predictor variables and constant.**

|  | $\beta$ (SE) | p | 95% CI for odds ratio | | |
|---|---|---|---|---|---|
|  |  |  | Lower | Odds ratio | Upper |
| Constant | 0.68 (0.24) | 0.01 |  |  |  |
| **Sex (male)** | **−0.71 (0.34)** | **0.03** | **0.25** | **0.49** | **0.94** |
| **Rearing (non-mother)** | 0.48 (0.35) | 0.17 | 0.82 | 1.62 | 3.29 |

Note:
Significant variables are bolded. $R^2 = 0.03$ (Hosmer-Lemeshow); 0.04 (Cox-Snell); 0.05 (Nagelkerke). Model $X^2(2) = 6.49$; $p = 0.04$.

3.18 times more likely to exhibit ABN-XC than those who were mother-reared. The other variables included did not contribute significantly to the model.

The best fit model for ABN-C included all three predictor variables and was statistically significant, $X^2(5) = 17.59$, $p < 0.01$. The regression coefficients and odds ratios are reported in Table 5. The model explained 13.6% (Nagelkerke's $R^2$) of the variance in the exhibition of coprophagy and correctly classified 67.3% of cases. The variable sex was negatively related to coprophagy such that male chimpanzees were 3.57 times less likely to exhibit coprophagy than female chimpanzees. The laboratory origin variable had a positive relationship with coprophagy such that chimpanzees that

**Table 4 Logistic regression model for non-coprophagy abnormal behavior (ABN-XC) with predictor variables and constant.**

| | $\beta$ (SE) | p | 95% CI for odds ratio | | |
| --- | --- | --- | --- | --- | --- |
| | | | Lower | Odds ratio | Upper |
| Constant | −0.45 (0.37) | 0.22 | | | |
| **Rearing (non-mother)** | **1.16 (0.44)** | **0.01** | **1.37** | **3.18** | **7.65** |
| Sex (male) | −0.09 (0.35) | 0.80 | 0.46 | 0.92 | 1.81 |
| Origin (lab) | 1.50 (0.90) | 0.09 | 0.89 | 4.50 | 34.5 |
| Origin (private) | −0.52 (0.78) | 0.50 | 0.13 | 0.60 | 2.76 |
| Origin (zoo) | −0.05 (0.43) | 0.91 | 0.41 | 0.95 | 2.26 |

Note:
Significant variables are bolded. $R^2$ = 0.08 (Hosmer-Lemeshow); 0.11 (Cox-Snell); 0.15 (Nagelkerke). Model $X^2(5)$ = 19.18; $p < 0.01$.

**Table 5 Logistic regression model for coprophagy (ABN-C) with predictor variables and constant.**

| | $\beta$ (SE) | p | 95% CI for odds ratio | | |
| --- | --- | --- | --- | --- | --- |
| | | | Lower | Odds ratio | Upper |
| Constant | −0.30 (0.38) | 0.38 | | | |
| **Sex (male)** | **−1.24 (0.37)** | **< 0.01** | **0.14** | **0.29** | **0.59** |
| **Origin (lab)** | **1.67 (0.76)** | **0.03** | **1.24** | **5.33** | **24.99** |
| Origin (private) | 1.17 (0.84) | 0.16 | 0.61 | 3.21 | 16.84 |
| Origin (zoo) | 0.63 (0.45) | 0.16 | 0.79 | 1.87 | 4.63 |
| Rearing (non-mother) | −0.88 (0.46) | 0.056 | 0.16 | 0.42 | 0.99 |

Notes:
Significant variables are bolded. $R^2$ = 0.08 (Hosmer-Lemeshow); 0.10 (Cox-Snell); 0.14 (Nagelkerke). Model $X^2(5)$ = 17.59; $p < 0.01$.

were born in a laboratory were 5.33 times more likely to exhibit coprophagy than those born in the wild.

## DISCUSSION

Our survey results revealed a lower prevalence of abnormal behavior in zoo-housed chimpanzees compared to the most recent published evaluation (*Birkett & Newton-Fisher, 2011*), which reported these behaviors as "endemic" and present in 100% of the zoo-housed subjects they sampled. Using similar categories of these behaviors in survey form, our data suggests that a lower prevalence of 64% of zoo-housed chimpanzees were observed to engage in some type of abnormal behavior over a two-year period. Although methodological differences may account for some of these differences, we assert that the current evaluation is a broader assessment of the prevalence of abnormal behavior in the zoo-housed chimpanzee population. The current study draws from a substantially larger sample of zoos (26 institutions compared to six) and subsequently surveys a broader range of individuals (165 subjects compared to 40). This is an important consideration given that, in this study and others, these behaviors are often linked to early rearing histories, which should be adequately represented in the study sample.

One clear similarity in the results of these two assessments is the prevalence of coprophagy as the most commonly reported abnormal behavior: 41% of the sample was reported to engage in this behavior. The link between this behavior and its utility as an indicator of wellbeing however, has recently been brought into question (*Nash et al., 1999*; *Hopper, Freeman & Ross, 2016*). There is growing evidence that coprophagy may be a socially-learned behavior and may not be as relevant an indicator of negative welfare as some other abnormal behaviors (*Nash et al., 1999*; *Hook et al., 2002*; *Freeman & Ross, 2014*; *Hopper, Freeman & Ross, 2016*). Though a socially-learned behavior could still be an indicator of negative welfare, it can likely be distinguished from those behaviors that are more directly tied to environmental or social inadequacies. Our analysis of the variables that predict the occurrence of coprophagy in this sample support this concept in a number of ways. First, we found a significant sex difference in the prevalence of coprophagy such that female chimpanzees were 3.57 times more likely than male chimpanzees to exhibit this behavior. This finding mirrors a past assessment (*Fritz et al., 1992*) and may be linked to sex differences in social learning. A study by *Lonsdorf (2005)* has directly demonstrated the biased proclivity of female offspring to be the recipients of socially-transmitted tool-using behavior and we argue that coprophagy may be learned similarly through cultural transmission (*Hopper, Freeman & Ross, 2016*).

Another finding that would support the idea that coprophagy is indeed a socially-transmitted behavior is a link to rearing history. Indeed we found that mother-reared chimpanzees were 2.38 times more likely than non-mother-reared chimpanzees to exhibit coprophagy, though this finding did not reach statistical significance. Mother-reared chimpanzees are likely to have more exposure to other chimpanzees who display coprophagy than those chimpanzees raised in a human setting (nursery or privately owned individuals typically live in smaller groups and their companions are less likely to show coprophagy), so the opportunity to learn this behavior socially may be heightened. In addition, mother-reared chimpanzees typically have more developed social skills than those raised in nurseries or by humans (*Spijkerman et al., 1997*; *Baker et al., 2000*; *Kalcher-Sommersguter et al., 2011*) which may allow them to better learn behaviors from their mothers and others in their social groups. For these two reasons we would expect these individuals to be more likely to learn a behavior like coprophagy. Also, given what we know about the negative welfare outcomes for non-mother-reared chimpanzees (*Fritz et al., 1992*; *Nash et al., 1999*; *Martin, 2002*; *Kalcher-Sommersguter et al., 2011*), if coprophagy was an indicator of negative welfare, we would expect to see these chimpanzees exhibit more of this behavior. Indeed the opposite trend is true in our sample, suggesting that coprophagy rates have relatively little to do with welfare. These results reinforce the established relationship between mother-rearing and elevated coprophagy (*Nash et al., 1999*; *Bloomsmith et al., 2006*; *Hopper, Freeman & Ross, 2016*) and the idea that the link between welfare and this behavior should be further evaluated.

Laboratory origin was also a significant predictor of coprophagy; chimpanzees born in research laboratory settings were 5.33 times more likely to exhibit coprophagy than

chimpanzees born in the wild. It is unclear why this relationship exists; further work is needed to compare the effects of different physical and management environments on the behavior of captive chimpanzees.

For some abnormal behaviors in primates we have empirical evidence to link the behaviors to suboptimal environments such as social isolation and non-mother-rearing (rocking: *Fritz et al., 1992*; self-injurious behavior: *Harlow & Harlow, 1965*; *Kraemer & Clarke, 1990*). These behaviors were reported less in this chimpanzee sample; rocking and self-injurious behavior were reported in fewer than 10% of chimpanzees. Overall, when we remove the occurrences of coprophagy, the prevalence of abnormal behaviors in the sample decreases to about half of the sample (48%). The most prevalent behavior after coprophagy was hair plucking (32%) which has been recognized as a relatively common abnormal behavior in many primate species (*Nash et al., 1999*; *Lutz, Well & Novak, 2003*; *Less, Kuhar & Lukas, 2013*; *Brand & Marchant, 2015*). However, the relationship between this behavior and psychological wellbeing is still unclear, as heredity and social learning may influence the expression of hair plucking (*Nash et al., 1999*; *Hook et al., 2002*; *Less, Kuhar & Lukas, 2013*). When examining the factors that influence the expression of these abnormal behaviors, we find some substantive differences from those factors influencing the expression of coprophagy.

As past studies have shown, rearing is associated with the occurrence of abnormal behaviors, although these studies have generally been correlational (*Harlow & Harlow, 1965*; *Rogers & Davenport, 1969*; *Fritz et al., 1992*; *Nash et al., 1999*; *Martin, 2002*). Our model supported this idea, demonstrating that chimpanzees who were not mother-reared were 3.18 times more likely to exhibit abnormal behavior (excluding coprophagy) than mother-reared chimpanzees. This rearing result is in the opposite direction as the trend revealed by our coprophagy-only model, again demonstrating differences in the ontogeny of these behaviors. We found no effect of sex, suggesting that in contrast to coprophagy, these behaviors are unlikely to be the result of social transmission and therefore may be more reliable indicators of individual welfare. The strong effect of rearing also emphasizes that welfare evaluation must consider the contribution of historical variables to present behavior, rather than solely focusing on proximate factors as the cause of all abnormal behavior.

Although this survey-based approach allows for a larger statistical sample, there are several potential methodological weaknesses which should be considered. One limitation of a survey method is that our results only show the prevalence of these behaviors without any information about their frequency or duration. As such, individuals who perform these abnormal behaviors on a daily basis cannot be distinguished from those who engage much more rarely, perhaps only once over a two-year period. Furthermore, we are also unable to determine the potential effect of other factors such as shifts in management protocols and social dynamics that may have influenced the expression of abnormal behaviors over that time period. Though more intensive observational studies could potentially address this weakness, they take considerably more time and resources to conduct adequately.

Another possible weakness is that these findings are based on reports from animal management staff working with the chimpanzees and may be vulnerable to subjective interpretation or even diminished opportunity to observe these behaviors. Although these are valid perspectives, we assert that even direct observations result in a relatively minute fraction of a chimpanzee's daily activities and that more generalized observations taken over the course of several years may be as likely to produce accurate prevalence estimates (*Whitham & Wielebnowski, 2009*; *Less et al., 2012*). Furthermore, the survey methods used here are ultimately a conservative measure, as abnormal behaviors need only be observed once in a two-year period, and are therefore likely to be *overestimating* the prevalence of these behaviors.

We assert that these survey data represent a useful evaluation of the prevalence of abnormal behaviors in zoo-housed chimpanzees. This study provides a broad characterization of the occurrence of abnormal behavior in zoo-housed chimpanzees and elucidates some of the variables in the life histories of chimpanzees that contribute to these behaviors. When considering our results and the effects of rearing, sex, and origin on the occurrence of coprophagy compared to the other abnormal behaviors, it is apparent that coprophagy, despite its prevalence, is associated differently with these factors. This supports previous proposals for coprophagy to be classified separately when assessing abnormal behavior and welfare of chimpanzees (*Hopper, Freeman & Ross, 2016*). Overall, this study calls into question *Birkett & Newton-Fisher's (2011)* assertion that abnormal behavior is pervasive in zoo-housed chimpanzees, but we join those authors in their support for work to assess and ultimately improve captive environments for chimpanzees. We acknowledge that we were unable to assess all of the many factors potentially associated with abnormal behaviors, and as a result, we encourage future research to include information on factors such as genetic relatedness, age, social exposure and more detailed early historical variables in order to refine our knowledge of these behaviors. Understanding not only the prevalence of abnormal behaviors but also focusing efforts on determining the causes of those behaviors most likely to inform us about chimpanzees' psychological states, should be a priority for managers and welfare scientists.

## ACKNOWLEDGEMENTS

We would like to thank all the zoo staff that completed surveys as part of this study. We would also like to thank the staff of the Fisher Center for their assistance in data analysis and manuscript preparation.

### Funding

This study was funded by the Leo S. Guthman Fund. The funders had no role in study design, data collection and analysis, decision to publish, or preparation of the manuscript.

### Competing Interests

The authors declare that they have no competing interests.

## Author Contributions

- Sarah L. Jacobson analyzed the data, contributed reagents/materials/analysis tools, wrote the paper, prepared figures and/or tables, reviewed drafts of the paper.
- Stephen R. Ross conceived and designed the experiments, performed the experiments, contributed reagents/materials/analysis tools, wrote the paper, reviewed drafts of the paper.
- Mollie A. Bloomsmith conceived and designed the experiments, wrote the paper, reviewed drafts of the paper.

## Data Deposition

The raw data has been supplied as Supplemental Dataset Files.

## Supplemental Information

Supplemental information for this article can be found online at http://dx.doi.org/10.7717/peerj.2225#supplemental-information.

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
