# Peer review of "Characterizing abnormal behavior in a large population of zoo-housed chimpanzees: prevalence and potential influencing factors"

_PeerJ, doi:10.7717/peerj.2225_

## Round 0.1 · original submission · Minor Revisions

· Academic Editor

Minor Revisions

First, thank you for submitting such important work to PeerJ. I was impressed by the large sample of zoological institutions surveyed and the concise, informative review of the issues involving identification of abnormal behaviors. I was fortunate to have two experts promptly review the MS and both have significant suggestions for improvement. Their comments are very clear so I will not reiterate them here, but I would like to emphasize my agreement with Reviewer 2 that it is important to avoid making causal statements based on correlational rather than experimental data. Obviously it is not ethically possible or desirable to manipulate rearing conditions for chimpanzees, but we must refrain from discussing etiology when we cannot know what the causal factors are, even when we observe strong correlations. For example, it is possible that infants rejected by their mothers, are rejected because they are less healthy in some way, and that this issue leads them both to be human-reared and to display more abnormal behaviors in the future. I’m not arguing that this is plausible but it needs to be addressed given the non-experimental nature of the predictors.

I also wish that there was a strong justification of the classification of abnormal behavior. We simply do not have the same regularity of data from wild chimpanzees or even home-reared chimpanzees with which to be certain of the abnormality of behaviors (or even opportunity to engage in some of these behaviors). I'd also like to see expanded discussion of the points regarding the extent to which 'abnormal' behaviors may be adaptive in abnormal environments. You allude to this but more could be said here. Abnormal behaviors could in fact be seen as innovations or indicators of behavioral flexibility (of course when pitted against self-harm). It would have been nice to have comparison data from caretakers at laboratory facilities, given that an important impetus for the paper seems to be discussion of the extent to which zoos evoke such behavior patterns. It would also be nice to have longitudinal data from the same individuals particularly with changes in their circumstances. I am mostly concerned with the definition of abnormal as a behavior "having occurred at least once in the past two years". This seems far too liberal, especially given the fallibility of human recollection. A stronger study would collect observations over perhaps a shorter period of time noting current conditions. As you point out, the study is limited in its inability to distinguish between frequent displays of abnormal behavior and rare occurrences. It is also unable to determine current factors that promote the expression of abnormal behaviors, which should be included as a limitation as well. I would therefore tone down the final paragraph.

I feel that one very important missing variable is the extent to which the chimpanzee had been socialized when young (or at all). Number of years in isolation may be an important variable to consider. If not mother-reared, did the chimpanzee have access to other chimpanzees, or playmates? We know from Harlow’s work how important this is.

Perhaps a note on why you wished to include only individuals from AZA accredited organizations would be helpful. Make it explicit.
It isn’t clear to me what the unit of analysis is – it doesn’t seem to be individual chimpanzees as you indicate the “proportion of individuals” – then was institution your unit of analysis? Please clarify.

Please avoid using “while” and “since” unless in a temporal sense.

Although you cannot change the methodology post-hoc, please revise the MS to be more conservative in line with the paper's limitations and emphasize the rationale and strengths further.
Please note Reviewer 1 has added comments directly to the attached PDF of your paper.

Reviewer 1 ·

Basic reporting

With respect to the English usage in the manuscript, there is some evidence of cumbersome sentence structure, run-on sentences, and either inappropriate or lack of comma use throughout the manuscript. A thorough reading by another individual with expertise in grammar and scientific writing would help sharpen the writing. Overall, the archival literature is well-presented in the Introduction and Discussion, and provide the necessary context for the paper. The paper appears to adhere to the standards of PeerJ, and the raw data have been included. Not all figures may be necessary, however, as Figures 5 & 6 are quite complex, busy, and redundant, as most of the findings relevant to most readers are presented in the text.

Experimental design

There are concerns with the experimental design as it pertains to some of the statements made throughout the paper, particularly in the Discussion section, where the authors have made several bold statements about the greater significance of their findings over a recent empirical, data-driven study that addressed some of the same questions. I have noted in a number of places in the manuscript, through the use of the Sticky Note option, where such statements have been made, and my reaction to them. Because both papers have such different approaches, the comparisons of the results between them are extremely limited. Please see marked manuscript document for further details.

Validity of the findings

Given that the manuscript is based on survey data, which can have built-in bias and subjectivity, as well as limited opportunity for observations by the caretakers (as noted by the authors), the findings may have limited validity. The responses tallied are based on subjective survey responses made by zookeepers and curators that may not represent a valid distribution of the behaviors in question. The authors appear to have tried to make the best of these findings, but need to be very cautious when making comparisons with previous published findings that have very differing methodologies and results.

Additional comments

Please see all the Sticky Notes within the body of the manuscript that were made during the review process. Any highlighted word or phrases that do not have an accompanying Sticky Note explanation are words that should be deleted.

Annotated reviews are not available for download in order to protect the identity of reviewers who chose to remain anonymous.

·

Basic reporting

This paper is very clearly written, well researched, and appropriately referenced. It makes a useful contribution to the literature as well.

Experimental design

The paper addresses a clearly defined, meaningful research question. Although the methodology (an institutional survey) is clearly appropriate for answering questions about the prevalence of abnormal behaviors in captive chimpanzees, it is less appropriate for answering questions about etiology, i.e. causality, which can only be determined by a controlled experiment. Therefore, it would be more appropriate for the authors to characterize their data as representing prevalence of and factors associated with the occurrence of abnormal behaviors, rather than etiology.

Some aspects of the experimental design are yet unclear and should be clarified or examined.

Line 152-153: What was the justification for including animals that had potentially only showed an abnormal behavior once in a two-year period? Is this consistent with other studies or was it just chosen as the most conservative approach? This seems like it could really overestimate the prevalence of abnormal behavior by including events that could be singular/incidental or potentially not meaningful.

Line 173-174: More information should be provided on the age distribution of the sample in addition to just the range. Also, why was age not taken into account in data analysis? Much of the discussion concerns the likelihood that abnormal behaviors are influenced by social learning, and at 2 years of age, would a chimpanzee necessarily have learned this behavior yet? Is there any literature on the ontogeny of abnormal behavior that would clarify at what age these behaviors are likely to appear?

Line 181-184, Figures 2a/b: Can these figures be combined to show to what extent the different rearing types were represented in each origin category? Was the interaction between these factors taken into account in statistical models? Also, perhaps some information here about the age distribution of each subgroup would be appropriate.

Line 198: I understand that wild was likely chosen as the reference variable for origin because it represents natural conditions, but would zoo perhaps be a more appropriate reference because it is the more common condition? Plus, it seems likely that origin would be conflated with age for wild-caught animals, since I am assuming these individuals entered the zoo population earlier when harvesting animals from the wild was more common.

Finally, did the authors take into account genetic relatedness or shared rearing experience in their analysis? Were the subjects that came from labs related to one another, or are there any other familial patterns that may need to be accounted for?

Validity of the findings

This study presents some compelling results that clearly benefit the body of knowledge on abnormal behaviors in chipmanzees, particularly with regard to clarifying the classification of coprophagy as an abnormal behavior. There are again some points that could be clarified here.

Line 259: Characterization of 64% as a “modest” prevalence is questionable. Although this is a lot less than the 100% from the Birkett and Newton-Fisher (2011) study, it’s still a solid majority of the zoo population, which is a lot!

Line 272: Although the attempt to directly compare the current dataset to that of Birkett and Newton-Fisher (2011) is a good idea, I’m not sure how meaningful the comparison is when it only includes 8 out of 40 of the original subjects. The direct comparison could be de-emphasized.

Line 281-283: Given the evidence that other abnormal behaviors, such as hair-plucking, may be socially transmitted, I am not sure it is logically sound to argue that coprophagy may not be relevant as an indicator for negative welfare simply because it may be socially learned. A behavior could be socially learned and still have a negative impact on welfare, the two constructs are not mutually exclusive.

Line 312-317: I think this section needs to be considered further. I’m assuming many of the laboratory chimps may not have been mother-reared, but this is where it would be helpful to show rearing*origin in the methods. Is the interaction (or lack thereof) referred to here between origin and rearing presented anywhere in the results section?

Line 322: Should this read chimpanzees rather than respondents?

Line 327-331: Again the assumption because a behavior is socially learned, it may not be indicative of compromised welfare seems unsubstantiated. Furthermore, it may be difficult (if not impossible) to separate out the impacts of social learning versus heredity in the expression of behaviors like hair-plucking, which should be taken into consideration when considering potential etiologies for these behaviors.

Line 371: I suppose it depends on how you define “pervasive’, but if 64% of animals in the study performed an abnormal behavior, it seems like an overly strong statement to claim that these data refute the assertion that abnormal behavior is pervasive. 64% of the population is still pretty prevalent, if not pervasive.

---

## Round 0.2 · Minor Revisions

· Academic Editor

Minor Revisions

Thank you for being responsive to the last round of reviews. I have a few very minor revisions for you before I can officially accept your MS.
On line 95, please replace "&" with "and"
On line 175-176, please provide a page # for this direct quote. Or, if not possible, due to online publication, paraphrase.
I now understand your comment that your measure of frequency is an overestimate (line 211). It will be important to note that the estimates refer to number of chimpanzees that expressed an abnormal behavior in a two year period, but that estimates really can't speak to the frequency of such behaviors in a population. I think it's really important to make this distinction. Accordingly, you might rephrase “prevalence of abnormal behavior in this population” (line 346); although you are using the term prevalence correctly, it can read as frequency to a novice reader. You include a nice discussion of this point on lines 468-476, you still use the term “prevalence” (also on line 519). Perhaps just be more specific here, that is by saying “the number of individuals who expressed this behavior at least once over the two year period”. Adding to the confusion, you state that the survey methods lead to an underestimate of prevalence (line 285). Perhaps some discussion as to the importance of prevalence, relative to frequency, is warranted in the conclusions.
"Figures” 1, 2, 4,5 are Tables, not figures. Please renumber all tables and figures accordingly, including the call-outs within the text. Please remove vertical lines from tables.
There are several missing commas, including after "rearing" on line 219, and after the ‘)’ on line 412. Please add commas between clauses throughout.
Should Ross (2015) also be cited on line 246?
The number of individuals in Figures (now Tables) 2a and 2b sum to 105, but you say you have data from 165 chimpanzees. Please clarify and make consistent. Also you could combine into one table that separates chimpanzees from different origins into mother-reared and non-mother reared categories, which would be more informative.
On line 269, change "effect on" to "relationship with" or "association with".
I don't think you need the % reported atop the bars on Figure 3 (now Fig. 1). It is evident enough from the graph itself what the figures represent.
I think it is important to control for age when looking at origin as a predictor given the high potential for age to be a confound here (this was also noted by one of the reviewers in the previous round of comments).
When referring to your studied chimpanzees, you should use the term “sample” not “population” as the latter term should be reserved for reference to captive chimpanzees in general (e.g. lines 418, 437, 439).
On line 448, change “affects” to “is associated with”.
Be consistent about placing references in chronological order within parentheses (lines 482-483).

Thank you once again for submitting such important work to PeerJ.

---

## Round 0.3 · accepted · Accept

· Academic Editor

Accept

Thank you for taking care of the requested revisions so promptly. I think these changes have helped to clarify the main take-home message of the paper.

Although I will not insist upon it here, I would strongly encourage you to include age as a controlled predictor in any further analyses you would conduct to examine the role of rearing and origin in chimpanzee behavior.